# Insights into de-mixing and morphology modulation in coacervate-membrane interactions from integrating experiments and simulations

Sayantan Mondal[1,7], Agustín Mangiarotti [2,3,6,7], Rumiana Dimova [2] & Qiang Cui [1,4,5]

Intrinsically disordered proteins can undergo liquid-liquid phase separation to form condensates or coacervates, which play numerous regulatory roles in the cell. Recently, it was recognized that membrane-adsorbed condensates are crucial for biomolecular localization, and in some cases, induce significant changes in membrane morphology. A detailed understanding of the underlying mechanisms remains incomplete. Here, we combine experiments and simulations to unravel structural and dynamic features of the coacervate/membrane interface across scales. We study poly-Lysine/poly-Aspartate ($K_{10}/D_{10}$) coacervates with different unilamellar liposomes. By combining confocal microscopy, hyperspectral imaging, fluorescence recovery after photobleaching, and two complementary coarse-grained approaches, we show that the membrane affinity of the $K_{10}/D_{10}$ coacervate can be tuned by the anionic lipid content and quantified through the intrinsic contact angle, both in vitro and in silico. We find that the membrane in contact with the condensate displays a nearly two-fold reduced fluidity compared to the bare membrane. This is attributed to the crowding of lipids at the contact region, resulting in decreased area per lipid. Moreover, we observed local lipid de-mixing upon coacervate adsorption. This study provides an effective framework for integrating experiment and computation to characterize the properties of coacervate-membrane interfaces that are critical to the functional impacts of these interactions.

The liquid-liquid phase separation (LLPS) of proteins and genetic material leads to the formation of biomolecular condensates[1]. These membraneless organelles (such as nucleolus, stress granules, processing bodies, etc.) provide an additional means to compartmentalize subcellular processes[2] and are involved in diverse functions, ranging from genome reorganization and transcription[3–5] to stress response[6,7] and signal transduction[8]. During the past decade, there has been an increasing interest in comprehending the formation, internal structure, and dynamical behavior of bio-molecular condensates, both experimentally and theoretically[9–21].

The intrinsically disordered regions (IDRs) in proteins and RNAs have been identified as the participants and key regulators of LLPS in biological systems. They can establish multivalent interactions, either homotypic or

heterotypic, leading to LLPS[22,23]. In this context, complex coacervates of polyelectrolytes/polyampholytes assembled by associative phase separation are used as models of phase-separated droplets because they capture key features of membraneless organelles and allow the development of biomimetic systems for synthetic biology applications[24–27].

In recent years, several biological processes involving the interaction of biomolecular condensates and membrane-bound organelles have been described[28–32]. This phenomenon, known as wetting[33–35], can promote the mutual remodeling of membranes and condensates[36–42], leading to a variety of elastocapillary phenomena[43,44]. In this regard, experiments in vitro and in silico with model systems permit the precise control and tuning of the physicochemical properties of both membranes and condensates allowing

[1]Department of Chemistry, Boston University, Boston, MA, USA. [2]Max Planck Institute of Colloids and Interfaces, Science Park Golm, Potsdam, Germany. [3]Centro de Investigaciones en Química Biológica de Córdoba, CONICET, Córdoba, Argentina. [4]Department of Physics, Boston University, Boston, MA, USA. [5]Department of Biomedical Engineering, Boston University, Boston, MA, USA. [6]Present address: Departamento de Química Biológica Ranwel Caputto, Universidad Nacional de Córdoba, Córdoba, Argentina. [7]These authors contributed equally: Sayantan Mondal, Agustín Mangiarotti. ✉e-mail: rumiana.dimova@mpikg.mpg.de; qiangcui@bu.edu

for visualization and quantification of the resulting morphologies. When in contact with giant unilamellar vesicles[45,46], lipid-anchored proteins can undergo two-dimensional phase separation at the membrane surface, leading to inward tubulation in vesicles due to compressive forces[38]. On the other side, non-anchored condensates undergo wetting transitions upon contact with membranes, resulting in mutual remodeling of the membrane and condensate[33,36,47,48]. In the presence of excess membrane area, the condensate-membrane interface can become ruffled, forming finger-like protrusions, indicating that condensates can promote complex membrane remodeling[48]. In addition, recent studies have shown that membrane wetting by biomolecular condensates can facilitate the endocytosis of the droplets[49–52], as well as influence the kinetic and thermodynamic coupling of the lipid and condensate phases[53,54]. This interaction also influences lipid packing and hydration, even to the extent of inducing phase separation in the membrane[48,50,55].

At the molecular scale, contact with a membrane can influence the conformational ensemble of intrinsically disordered proteins (IDPs), potentially altering their phase behavior. For example, fused in sarcoma forms fibrillar β-sheet rich structures on phosphatidylserine membranes, whereas it forms entangled condensate on a phosphatidylglycerol membrane[47,56]. Hence, predicting IDP phase behavior upon membrane wetting cannot be solely based on increased local concentration. Furthermore, the phase behavior of membranes and IDPs can be coupled[53,57,58]. Membranes can promote LLPS at their surface, even under conditions unfavorable for LLPS in bulk solutions[59,60]. These studies highlight the importance of membrane surfaces as critical regulators of LLPS and condensate properties[39,61]. Additionally, LLPS provide a novel mechanism for membrane remodeling in cells, complementing established mechanisms such as hydrophobic insertion, scaffolding, protein crowding, and coating[62–64].

Interaction between polyelectrolyte/polyampholyte coacervates and membranes has been shown to lead to significant membrane remodeling. For example, polyelectrolyte coacervates made of an equimolecular mixture of $E_{30}$ and $K_{30}$ can induce a pronounced negative curvature and local lipid de-mixing when partially wetting an anionic lipid bilayer[36]. Further studies demonstrate that the extent of such membrane remodeling depends on the charge patterning of the IDR chains[40]. Continuum mechanics-based descriptions suggest that the increased inter-protein contacts, driven by negative curvature, are the primary drivers of the observed membrane remodeling[38]. However, recent evidence demonstrates that such hypothesis may not hold in case of coacervate-membrane interaction where counterions play an important role in providing enthalpic stabilization[36].

Although several studies have explored specific aspects of the interaction of polyelectrolyte coacervates with membranes using either in vitro or in silico approaches[36,37,49,50,65], a systematic and integrative investigation exploiting the strengths of both methodologies is lacking. Here, we bridge this gap by combining experiments with molecular dynamics, providing a multiscale approach to unravel the effects of coacervate wetting on membrane properties and remodeling.

## Materials and methods
### Experimental procedures
**Materials.** The phospholipids 1,2-dioleoyl-sn-glycero-3-phosphocholine (DOPC) and 1,2-dioleoyl-sn-glycero-3-phospho-L-serine (DOPS) were purchased from Avanti Polar Lipids (IL, USA). ATTO 647N-DOPE was obtained from ATTO-TEC GmbH (Siegen, Germany). Chloroform HPLC grade (99.8%) was purchased from Merck (Darmstadt, Germany). All lipid stocks were prepared in chloroform at 4 mM, containing 0.1 mol % ATTO 647N-DOPE, and stored until use at −20 °C. Sodium chloride (NaCl), potassium chloride (KCl) and magnesium chloride (MgCl₂) were obtained from Sigma-Aldrich (Missouri, USA).

The oligopeptides poly(l-lysine hydrochloride) ($K_{10}$) and poly(l-aspartic acid sodium salt) ($D_{10}$), each with a degree of polymerization $n = 10$, were purchased from Alamanda Polymers (AL, USA) and used without further purification (purity ≥ 95%). The N-terminal TAMRA-K10 was

purchased from Biomatik (Ontario, Canada). All solutions were prepared using ultrapure water from SG water purification system (Ultrapure Integra UV plus, SG Wasseraufbereitung) with a resistivity of 18.2 MΩ cm.

**Condensate formation.** Phase separation was induced by mixing equimolar aliquots of K10 and D10 (5 mM) in a solution containing 15 mM KCl, 0.5 mM MgCl₂, and 170 mM glucose. For labeling, TAMRA-$K_{10}$ was added at a concentration of 1 mol%. The final osmolarity of the mixture was ≈ 200 mOsm which was adjusted using a freezing point osmometer (Osmomat 3000, Gonotec).

**Vesicle preparation.** Giant unilamellar vesicles (GUVs) were grown using the electroformation method. Briefly, 3–4 μL of the lipid stock were spread on two conductive indium tin oxide-coated glasses and kept under vacuum for 1 h. A chamber was formed using a rectangular 2-mm-thick Teflon frame sandwiched between the two glass electrodes. The lipid films were hydrated with 2 ml of an isotonic sucrose solution (matching the osmolarity of the condensate solution). An electric AC field (1 V, 10 Hz, sinusoidal wave) was applied for 1 h at room temperature. Once formed, the suspension of the giant vesicles was stored at room temperature until use. The vesicles were prepared freshly before each experiment.

**Condensate-membrane suspensions.** For the interaction of membranes with $K_{10}/D_{10}$ condensates, the vesicle suspension was diluted 1:10 into the final buffer matching the droplets suspension conditions. Then, an aliquot of this solution was mixed with the droplet suspension at an 8:1 volume ratio directly on a coverslip. The sample was then sealed for immediate observation under the microscope.

The GUVs were not immobilized on the coverslip through specific interactions but rather settled by gravity. This approach was chosen to avoid vesicle adhesion, which could alter the GUV geometry and thereby affect the accuracy of the contact angle measurement. To prevent the adhesion of GUVs to the glass surface, as well as the wetting of the coverslip by the condensates, we cleaned the coverslips with EtOH and water and then passivated them with a 2.5 mg/mL BSA solution. To facilitate stabilization by gravity, and thus imaging, vesicles are filled with a sucrose solution and then diluted in the (isotonic) condensate buffer containing glucose; this makes vesicles sink to the bottom of the chamber due to the density difference between the vesicle interior and exterior. In addition, the vesicles were further immobilized by the interaction with the condensate dense phase, which facilitated imaging as well as fluorescence recovery after photobleaching (FRAP) experiments. Figure S10 confirms that condensates do not wet the glass and remain spherical. When interacting with vesicles, condensates wet the membrane and deform the membrane-condensate interface.

**Confocal microscopy and FRAP.** A Leica SP8 confocal microscope equipped with a 63×, 1.2 NA water immersion objective (Mannheim, Germany) was used for imaging. TAMRA-K10 and ATTO 647N-DOPE were excited using the 561 nm and 633 nm laser lines, respectively. For FRAP measurements, a circular region of interest (ROI) with a diameter of 2 μm was chosen at the vesicle equator and photobleached during 3 iterative pulses over a total duration of ~3 s. The fluorescence recovery within the ROI was recorded and analyzed using ImageJ.

The apparent diffusion constant can be obtained by the FRAP measurements through Eq. (1)

$$D_{app} = \frac{r_0^2 v}{4 t_{\frac{1}{2}}}$$ (1)

where $r_0$ is the radius of the bleaching spot and $v$ is a correction factor accounting for the difference between the defined size of bleaching spot and its real size[45]. As the measurements conditions were the same for the wetted and bare parts of the vesicles (as well as for the different vesicles), the ratio of

$t_{1/2}$ for the wetted and bare parts of the membrane becomes equal to the ratio of the diffusion coefficients [Eq. (2)]:

$$\frac{t_{\frac{1}{2}}wet}{t_{\frac{1}{2}}bare} = \frac{D_{app}bare}{D_{app}wet} \tag{2}$$

Since we compare ratios of diffusion coefficients rather than absolute values, the geometric and methodological constraints specific to each measurement are expected to largely cancel out.

**Contact angle determination.** Confocal projections enable the determination of three apparent microscopic contact angles at the contact line between the vesicle and the condensate droplet, $\theta_e$ (opening toward the external solution, $\theta_c$ (toward the condensate phase), and $\theta_v$ (toward the vesicle interior). Accurate measurements of these contact angles between the different surfaces of the bare and wetted membrane segments and the condensate interface, require that the rotational axis of symmetry of the vesicle-droplet system lies in the imaging plane. This alignment is crucial to ensure precise geometry and contact angle determination, as explained in detail in ref. 37. Misalignment can lead to incorrect assessment of the system geometry and contact angles.

The geometric factor characterizing the membrane wetting affinity is defined as[37,66,67]

$$\Phi = \frac{\sin\theta_e - \sin\theta_c}{\sin\theta_v} \tag{3}$$

$\Phi$ can vary from $+1$ (a de-wetted state) to $-1$ (complete wetting). At the nanometer scale, the membrane is smoothly curved[68], and the wetting is characterized by the intrinsic contact angle ($\theta_{in}$). The intrinsic contact angle, which is contrary to the apparent ones, is a material property related to the geometric factor through: $\Phi = \cos\theta_{in}$[69].

**Spectral phasor analysis.** Hyperspectral images of vesicles labeled with LAURDAN were analyzed using the spectral phasor method[70]. This analysis calculates the real ($G$) and the imaginary ($S$) component of the Fourier transform of the spectra. The Cartesian coordinates ($G$ and $S$) of the spectral phasor plot are defined by the following expressions given by Eqs. (4) and (5).

$$G = \frac{\int_{\lambda_{min}}^{\lambda_{max}} d\lambda\, I(\lambda) \cos\frac{2\pi n(\lambda - \lambda_i)}{\lambda_{max} - \lambda_{min}}}{\int_{\lambda_{min}}^{\lambda_{max}} d\lambda\, I(\lambda)} \tag{4}$$

$$S = \frac{\int_{\lambda_{min}}^{\lambda_{max}} d\lambda\, I(\lambda) \sin\frac{2\pi n(\lambda - \lambda_i)}{\lambda_{max} - \lambda_{min}}}{\int_{\lambda_{min}}^{\lambda_{max}} d\lambda\, I(\lambda)} \tag{5}$$

where, for a given pixel, $I(\lambda)$ is the pixel intensity at wavelength $\lambda$, $n$ denotes the harmonic number, and $\lambda_i$ is the initial wavelength.

The spectral phasor approach follows the rules of vector algebra, known as the linear combination of phasors. This property implies that a combination of two independent fluorescent species (or states) will appear on the phasor plot at a position that is a linear combination of the phasor positions of the two independent spectral species. In this manner, two-cursor analysis was used to calculate the histogram for the pixel distribution along the trajectory for changes of LAURDAN fluorescence. The histograms represent the number of pixels at each step along the line between two cursors, normalized by the total number of pixels. We plotted the average value for each histogram ± standard deviation, as well as the center of mass of the histogram for quantitative analysis with descriptive statistics. The

center of mass was calculated following Eq.(6) as:

$$CM = \sum_{i=0}^{i=1} i\, F_i \Big/ \sum_{i=0}^{i=1} F_i \tag{6}$$

where $F_i$ is the fluidity fraction.

**Statistics and reproducibility.** At least three independent experiments were used for statistical analysis. In hyperspectral imaging experiments, the pixel histograms are shown as means ± standard deviation (SD), and the center of mass is represented as individual measurements together with the mean values ± SD. We employed One-way ANOVA and Tukey post-test analysis ($p < 0.0001$, **** | $p < 0.001$, *** | $p < 0.01$, ** | $p < 0.05$, * | ns = non-significant) for statistics. All statistical analyzes and data processing were performed with the Origin Pro software (Originlab corporation). All the data and microscopy images shown are representative of at least three independent experiments.

## Theoretical and computational procedures
**Computational modeling.** Because of the vast conformational space of IDRs, molecular dynamics (MD) simulations in the atomistic resolution (with explicit molecular water and ions) are computationally expensive. Therefore, one often resorts to different levels of coarse-grained (CG) approaches[71,72], for example, single-bead per amino-acid residue, multiple-beads per residue, single-bead for multiple residues, etc. Here, we have used CG modeling of two different resolutions, namely, (i) a generic model that consists of Cooke three-bead lipids[73] with bead-spring polymers; and (ii) MARTINI 3 model with explicit water/ion beads[74]. The primary caveat of CG approaches is the loss of atomistic and dynamic information. However, CG models can efficiently provide structural information at biologically relevant length-scale and long timescale that are otherwise inaccessible to atomistic MD.

**The generic coarse-grained model.** The generic model consists of a liposome made of three-bead Cooke lipids (Fig. 1e, *right panel*), and two types of polymer chains ($A_{10}$ and $B_{10}$) in equimolecular proportions that represent the polyelectrolytes ($K_{10}/D_{10}$). This setup has no explicit beads for water and ions. Such models have been successful in recapitulating several experimentally observed phenomena such as coacervate engulfment and multilayered membrane formation[50]. The polymers are modeled as self-repulsive chains implemented through the Weeks–Chandler–Anderson (WCA) potential[75] with heterotypic attraction implemented through the Lennard-Jones (LJ) potential described by Eq. (7) and Eq. (8), respectively

$$U_{WCA} = 4\varepsilon \left[\frac{\sigma^{12}}{r^{12}} - \frac{\sigma^6}{r^6} + \frac{1}{4}\right] \text{ when } r \le r_c$$
$$= 0 \text{ when } r > r_c \tag{7}$$

and

$$U_{LJ} = 4\varepsilon \left[\frac{\sigma^{12}}{r^{12}} - \frac{\sigma^6}{r^6}\right] \text{ when } r \le r_c^{LJ}$$
$$= 0 \text{ when } r > r_c^{LJ} \tag{8}$$

where $r$ is the inter-bead distance, $\varepsilon$ is the well-depth, $r_c = 2^{1/6}\sigma$ and $r_c^{LJ} = 2.5\sigma$, and $\sigma$ is the bead diameter. There are harmonic bonds between successive beads in the same polymer with a dimensionless force constant of 50.0 and the equilibrium bond length $r_{eq} = 0.548$. The well depth $\varepsilon$ is set to be 1.5 between the polymer beads. $\sigma_{AA}$ and $\sigma_{BB}$ both are set to 0.5. All values for this model are in the reduced unit where $1\sigma \approx 6.93\text{Å}$.

Each lipid molecule consists of three beads, one for the head (H) and two for the tail (T). The heads are self-repulsive with WCA potential

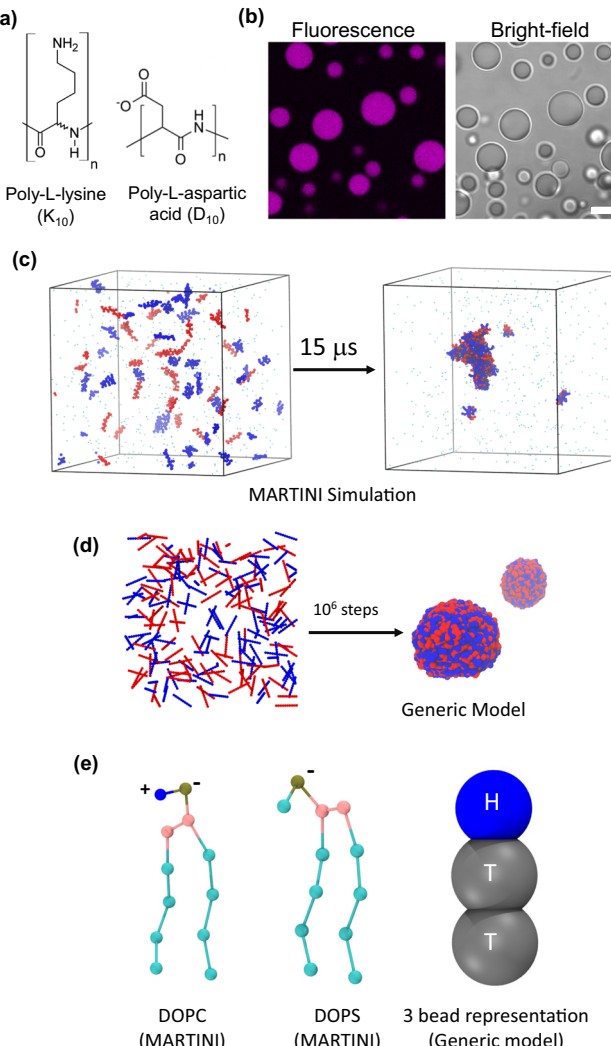

**Fig. 1 | Molecular structures, models and coacervate formation in experiments and simulations. a** Molecular structures of the poly-lysine ($K_{10}$) and poly-aspartic acid ($D_{10}$) oligopeptides. The number of monomers (n) is 10. **b** Confocal fluorescence microscopy and bright-field images of coacervates formed by the equimolar mixture of $K_{10}$ and $D_{10}$ peptides (~5 mM) in 15 mM KCl and 0.5 mM MgCl$_2$. A 1 mol% of labeled peptide ($K_{10}$-TAMRA) was included for visualization. Scale bar: 20 μm. **c** Coacervation of an equimolecular mixture of $K_{10}$ and $D_{10}$, as observed in MARTINI based coarse-grained simulation for 15 μs. **d** Coacervation observed from a generic CG model where the amino acids are represented by single beads ('A' and 'B') which are self-repulsive (A-A and B-B) but heterotypically attractive (A-B). **e** Lipid models used in this study: MARTINI lipids and the three-bead representation (H-T-T) for the generic CG model.

[Eq. (7)] and the tail beads are attractive according to a combination of truncated LJ and cosine potential, given by Eq. (9):

$$
\begin{aligned}
U_{TT} &= -\varepsilon \quad \text{when } r < r_c \\
&= -\varepsilon \cos^2 \frac{\pi(r - r_c)}{2\omega_c} \quad \text{when } r_c \leq r \leq (r_c + \omega_c) \\
&= 0 \quad \text{when } r > (r_c + \omega_c)
\end{aligned}
\tag{9}
$$

Here, $\sigma_{HH} = 0.95 = \sigma_{HT}$ and $\sigma_{TT} = 1.00$ (recall that $r_c^{LJ} = 2.5\sigma$). The $\omega_c$ parameter for TT, $\omega_{TT}$, could be different depending on the lipid composition and stiffness of the bilayer. The successive lipid beads are linked by finite extensible nonlinear elastic[76] bonds [Eq. (10)] with $k =$

30.0 and $r_\infty = 1.5$

$$
V_{FENE}(r) = -\frac{1}{2} k r_\infty^2 \log\left[1 - \left(\frac{r}{r_\infty}\right)^2\right]
\tag{10}
$$

The polymer-head (A-H and B-H) interactions are attractive LJ type, and the polymer-tail (A-T and B-T) interactions are repulsive WCA type. Therefore, we have two tunable parameters to describe different systems with the same model in a mean-field manner, namely, the polymer-head interaction strength ($\varepsilon_{PL}$) and the lipid inter-tail interaction range ($\omega_{TT}$). The rationale behind the choice of the two tunable parameters is the following. When the lipid composition in the vesicle changes, the stiffness (or, bending modulus) of the bilayer as well as the effective attraction between the lipid and polymers will change.

**MARTINI coarse-grained model.** As mentioned above, we model the system using another general-purpose, explicit-solvent, and popular CG force-field, MARTINI (v3.0.0)[74]. MARTINI uses a 4-to-1 mapping strategy where 4 non-hydrogen atoms are represented by one bead. MARTINI-3 was used previously to study the salt-concentration dependence of polyelectrolyte coacervation[77] and later to study coacervate-membrane interaction[36]. In the context of IDPs, a rebalancing strategy has been proposed by scaling the protein-water interactions in MARTINI-3 that improves the single-chain behavior like radius of gyration and end-to-end distance[78]. In addition, the MARTINI model is known to provide a reasonable description of the lipid bilayers[79]. In MARTINI, each D and K residue is modeled by two CG beads, one for the backbone and one for the sidechain. The sidechain bead of each D bears a −1e charge and the side-chain bead of each K bears a +1e charge (e=charge of a proton). The N- and C-termini, respectively, bear +1e and −1e charges. For the lipid bilayer, we have used two different kinds of lipids, namely, DOPC and DOPS, in different ratios. The details of parameterization can be found in the original MARTINI 3 paper[74].

**Molecular dynamics simulation details.** For the generic CG model, we first simulate 1000 copies of each polymer chain inside a (40 unit)$^3$ cubic box to form the droplet (Fig. 1d). A cluster analysis is provided in the Supporting Information (SI, Fig. S3). We separately prepare a unilamellar vesicle of 50 unit diameter with ~9500 three-bead lipid molecules inside a (70 unit)$^3$ cubic box. After that, the preformed droplet is placed on the outside of the vesicle. The systems are first energy minimized and then equilibrated for $10^5$ steps in an NVT (constant particle number, volume and temperature) ensemble with a timestep ($\Delta t^*$) of 0.002 (in reduced units), during which the initial adsorption occurred. Following this, we perform production runs for another $10^7$ steps in an NVT ensemble with $\Delta t^* = 0.005$. We use a Langevin thermostat to keep the average reduced temperature ($T^*$) at 1.1 with a damping timescale of 50.0. The coordinates are recorded in every $10^4$ steps. For these simulations, we used the LAMMPS[80] package and VMD[81] for visualization.

For the MARTINI simulations, the atomistic versions of the polyelectrolytes ($K_{10}$ and $D_{10}$) are prepared using *pymol*[82], followed by the conversion in MARTINI-CG format using the *martinize2* script[83]. Mimicking the experimental conditions, we first simulate 33 copies of each polyelectrolyte inside a (30 nm)$^3$ cubic box filled with 226,627 MARTINI water beads, 271 Na$^+$ and 271 Cl$^-$ ion beads. This is done for 15 μs to check the LLPS propensity in the bulk (Fig. 1c). A cluster analysis is provided in the SI (Fig. S4). Then we place the preformed droplet near different equilibrated lipid bilayer patches of dimension (32 nm)$^2$, each consisting of 3042 lipid molecules. However, the original MARTINI force field was unable to capture the experimentally observed adsorption of the coacervate on the DOPC bilayer and showed weak/transient adsorption with the 20% DOPS bilayer, suggesting that the interactions between polyelectrolytes and lipids are underestimated. Therefore, we re-scale some of the force field parameters by scaling up the interactions between

**Fig. 2 | Wetting and lipid redistribution in membrane-coacervate interactions. a** Confocal fluorescence images of DOPC or DOPC:DOPS GUVs labeled with 0.1 mol% ATTO 647N-DOPE (green) in contact with the TAMRA-labeled $K_{10}/D_{10}$ coacervates (magenta) at the indicated membrane compositions. NaCl was added to pure DOPC vesicles when indicated (first image). In all cases, the final concentrations of KCl and $MgCl_2$ were 15 mM and 0.5 mM, respectively. Scale bars: 5 μm. **b** Snapshots from the generic model molecular dynamics simulation at fixed value of $\varepsilon_{PP} = 1.5$ and different combinations of $\varepsilon_{PL}$ and $\omega_{TT}$ which correspond to the systems shown in panel a as indicated above the snapshots. **c** Observation from MARTINI simulations of $K_{10}/D_{10}$-coacervate in contact with bilayer patches of different compositions (pure DOPC, DOPC:DOPS = 9:1 and DOPC:DOPS = 8:2), which demonstrate wetting, negative membrane curvature generation, and local lipid de-mixing, respectively. Approximately 50% of the charged DOPS lipids (green) in the leaflet facing the coacervate accumulate in the contact area.

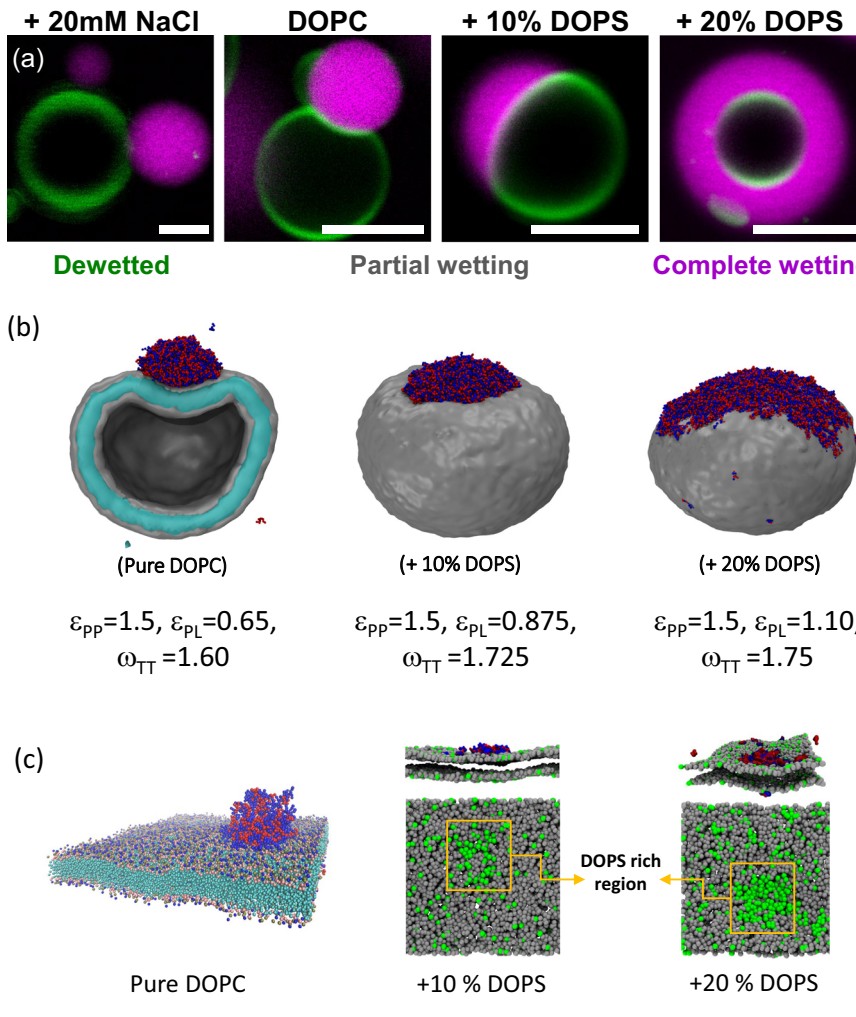

the polyelectrolytes and lipid heads. A similar technique was adapted earlier in the context of the membrane protein association[84]. Here we scale the non-bonded interaction strength between the protein beads and the lipid headgroup beads (four of them) of DOPC lipids by a factor of 1.20. On the other hand, due to the presence of Coulombic attraction between DOPS and polyelectrolytes, we do not scale the LJ parameters associated with DOPS charged head. In the SI, we provide a table of modified non-bonded interactions between the lipid heads and the polyelectrolytes (Table S1). A comparison of the density profiles of the lipids and proteins for both the unscaled and scaled forcefields are provided in the SI (Fig. S5). We first minimize the energy of the systems followed by 100 ns equilibration in an NpT ensemble ($T = 298$ K and $p = 1$ bar). After that a 5 μs production run is carried out with the same $T$ and $p$. All equilibration simulations are propagated with a time step of 10 fs and production simulations are propagated with a time step of 20 fs, using the leap-frog algorithm. We use the V-rescale thermostat ($\tau_T = 1 \text{ ps}^{-1}$) at 298 K and Parrinello–Rahman barostat with semi-isotropic pressure coupling ($\tau_P = 12 \text{ ps}^{-1}$) at 1 bar to make the bilayer tensionless. For initial equilibration purpose, we use Berendsen barostat with $\tau_P = 6 \text{ ps}^{-1}$. The electrostatic interactions are screened with a reaction-field ($\varepsilon_r$) of 15 within a cut-off of 1.1 nm and non-bonded interactions are also terminated at 1.1 nm with the Verlet cut-off scheme. The coordinate dumping rate is set to 100 ps. For all the MARTINI simulations, we use the GROMACS 2018.3 simulation package[85] and VMD[81] for visualization. For the calculation of the area per lipid (APL), we used the FATSLiM code[86] which uses local Voronoi cell construction

approach and is shown to be robust in tackling curved/deformed membrane surfaces.

## Results and discussions

### Membrane wetting and remodeling by peptide coacervates

We formed coacervates of the oligopeptides poly-L-lysine ($K_{10}$) and poly-L-aspartate ($D_{10}$) (see Fig. 1a) in a solution containing 15 mM KCl and 0.5 mM $MgCl_2$, as previously described[24,37]. Figure 1a shows the molecular structure of the polypeptides. By using a 1 mol% of TAMRA-K10, the coacervates can be observed with fluorescence confocal microscopy, as shown in Fig. 1b. In the simulations, the polyelectrolyte mixtures underwent LLPS with both MARTINI-3 (Fig. 1c) and the generic model (Fig. 1d) force fields.

To evaluate the coacervate interaction with membranes we used GUVs composed of pure DOPC and DOPC:DOPS mixtures, in the presence and absence of sodium chloride (NaCl). For observation by confocal fluorescence microscopy, the membranes were labeled with a 0.1 mol% of ATTO 647N-DOPE. Figure 2a shows that by tuning membrane charge and the salinity of the medium the interaction between membranes and the $K_{10}/D_{10}$ coacervates can be modulated. As previously shown for glycinin condensates[48], and other complex coacervates[37], two wetting transitions can occur, from de-wetting (that is, no interaction), to partial wetting and complete wetting (complete spreading of the droplet over the membrane surface). Wetting of the membrane by biomolecular condensates imply mutual remodeling, since the condensate spreads on the membrane when the affinity is increased, and the membrane gets locally reshaped[36,37,39,50].

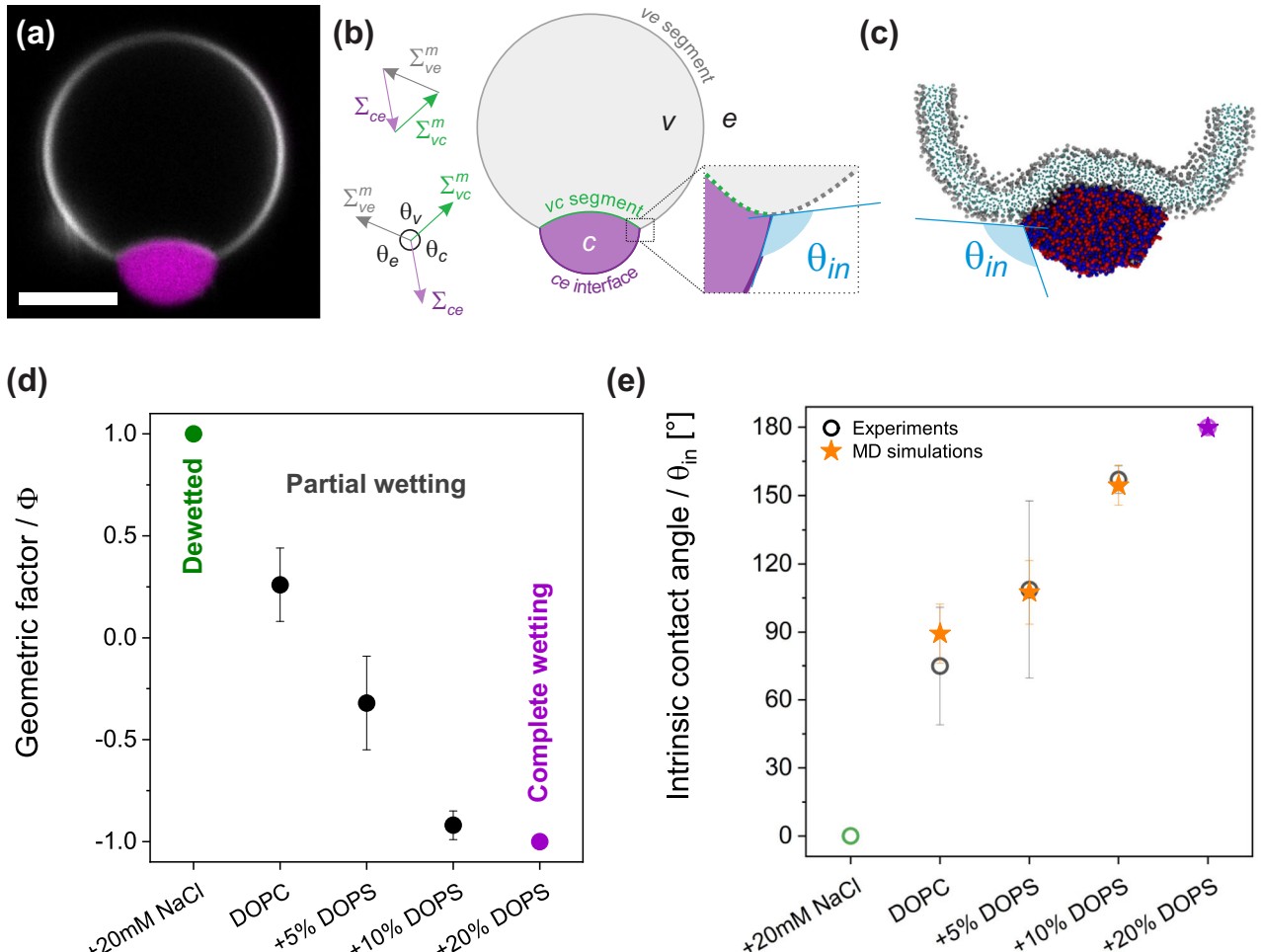

**Fig. 3 | Wetting morphologies and intrinsic contact angles in vesicle-coacervate systems. a** DOPC GUV in contact with a $K_{10}/D_{10}$ coacervate. Scale bar: 5 μm. **b** For partial wetting morphologies, the contact line between the condensate bare surface, *ce*, and the vesicle partitions the membrane into the *vc* and *ve* segments (green and gray, respectively), with the microscopic (apparent) contact angles $\theta_e + \theta_c + \theta_v = 360^o$. The droplet interfacial tension $\Sigma_{ce}$ and the mechanical tensions $\Sigma_{vc}^m$ and $\Sigma_{ve}^m$ of the two membrane segments are balanced and form the sides of a triangle. The apparent contact angles can be determined from the confocal cross-sections, and the geometric factor, $\Phi$, can be calculated according to Eq. 3, while the intrinsic contact angle is given by

$\Phi = \cos \theta_{in}$. **c** The intrinsic contact angle, $\theta_{in}$ as assessed from a snapshot from the generic model simulations. **d** Geometric factor, $\Phi$, experimentally estimated for vesicles of different membrane and milieu compositions. The system undergoes two wetting transitions, from dewetted ($\Phi = 1$) to partial wetting ($-1 < \Phi < 1$) and complete wetting ($\Phi = -1$) as a function of membrane surface charge and solution salinity. **e** Intrinsic contact angle $\theta_{in}$ assessed from the confocal microscopy images (hollow circles), for the systems shown in d and from simulation snapshots (orange stars) with the generic CG model and Cooke lipids. As the model does not include the effect of ions, we omit the leftmost point in simulations which represents complete dewetting.

In the generic CG model simulations using the generic Cooke lipid model, we explored the parameter space by keeping the interaction between peptides fixed at $\varepsilon_{PP} = 1.5$ and by tuning the peptide-lipid ($\varepsilon_{PL}$) and lipid-lipid interaction range ($\omega_{TT}$). The objective was to optimize these interaction parameters to match the geometric factor, i.e., the intrinsic contact angle, with experimentally measured values. Once refined, these parameters enable us to leverage the predictive power of the simulations to resolve the structural rearrangements in the bilayer and changes in lipid dynamics upon wetting by the coacervates.

The optimized parameters are shown below Fig. 2b. The three systems correspond to pure DOPC, 10% DOPS, and 20% DOPS compositions, respectively. In addition, we simulated another system that corresponds to 5% DOPS with $\varepsilon_{PL} = 0.7625$ and $\omega_{TT} = 1.70$ (not shown in Fig. 2b). The systems exhibit different extent of wetting. From MARTINI-3 simulations (Fig. 2c) we studied three systems, namely pure DOPC, 10% DOPS, and 20% DOPS to probe the spatial distribution of lipids upon wetting by the coacervate. We observed that the coacervate induced local de-mixing around the area of contact and generated a negative curvature (from lumen's perspective) on the 10% DOPS and 20% DOPS membrane (Fig. 2c *middle and right panels*). We further quantified that almost 50% of the anionic

lipids (DOPS) in the bilayer leaflet facing the coacervate are in contact with it, indicating a strong sorting and local de-mixing of the membrane. Our data shows that Lys (K) residues form more contact with the lipid heads, particularly with DOPS. A detailed analysis of the number of contacts between the components of polyelectrolytes and lipids is provided in SI (Fig. S8a, b). We find that the wettability of the lipid bilayers by the coacervate increases with increasing DOPS content (Fig. S8c). This local de-mixing and remodeling of the membrane could be crucial for processes like signaling and information transduction across the bilayer[8,87]. Moreover, it has been shown that the electric potential gradient generated by the condensate formation can influence the membrane potential, offering additional means for membrane regulation[14].

### Quantifying the membrane-coacervate interaction at the micro- and nanoscales

At the micron scale, the vesicle membrane displays a "kink" at the contact line between the membrane, the coacervate phase, and the external solution, as illustrated in the partial wetting examples shown in Figs. 2a and 3a. This kink cannot persist at the nanometer scale, as it would imply infinite bending energy. Contact angles measured from microscopy reflect the

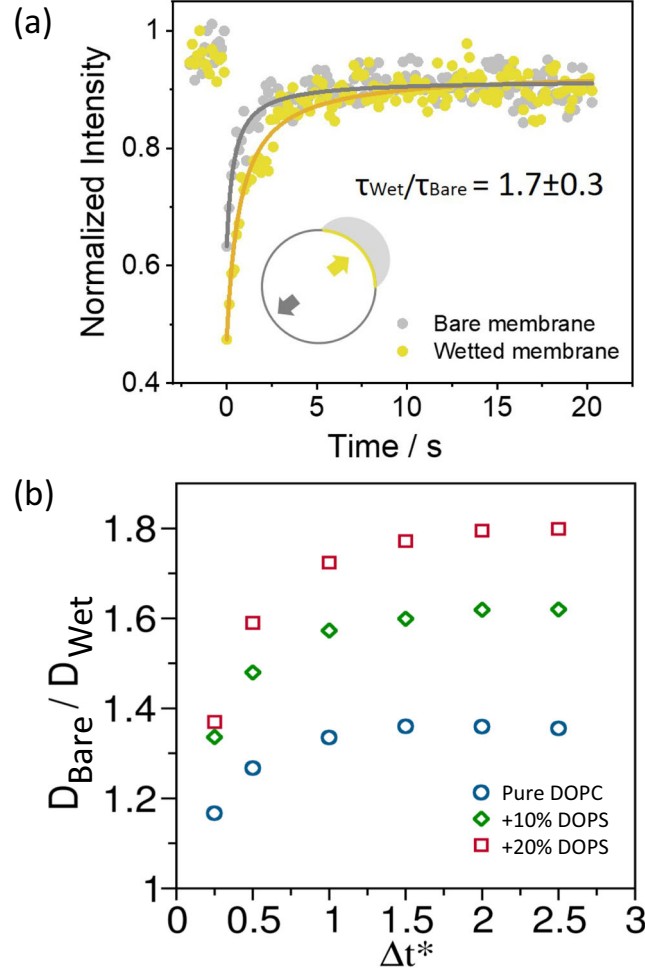

**Fig. 4 | Lipid diffusion in the membrane region wetted by the coacervate is slower than in the bare membrane. a** Fluorescence recovery after photobleaching (FRAP) of the fluorescent probe ATTO 647N-DOPE in DOPC:DOPS (9:1) membrane segments wetted by the coacervate (yellow) and the bare membrane (gray). Curves were fitted to the function: $y = (I_0 + I_{max}(x/\tau_{1/2}))/(1 + x/\tau_{1/2})$, where $I_0$ is the initial intensity, $I_{max}$ is the maximal intensity and $\tau_{1/2}$ is the halftime of recovery. **b** Ratio of lipid diffusion constants between the wetted and the bare membrane regions as obtained from the generic model CG simulations for the three systems (see Fig. 2b).

overall geometry of the vesicle-condensate system and depend on their areas and volumes. Under identical conditions, vesicle-condensate pairs in the same sample may exhibit different microscopic contact angles. However, at the nanometer scale, the membrane must exhibit a smooth curvature (see Fig. 3b), forming an intrinsic contact angle with the condensate interface, as shown previously[69,88]. The smooth curvature of the membrane has been also experimentally confirmed using super-resolution microscopy[68] and is also evident in the simulation snapshots, see Fig. 3c.

The intrinsic contact angle is a material property, unlike the microscopic contact angles, and is uniform for all condensate-vesicle pairs in the sample under fixed conditions[37,44,66]. At the three-phase contact line, the force balance between the condensate interfacial tension ($\Sigma_{ce}$) and the tensions of the two vesicle membrane segments ($\Sigma_{vc}^m$ and $\Sigma_{ve}^m$) forms a triangle (analogous to the Neumann triangle)[67], see Fig. 3b. We refer to these microscopically measured contact angles as "apparent" because they do not persist at the nanoscale. By considering the force balance and that the membrane has different affinities for the condensate and external phase, the intrinsic contact angle can be estimated from measurements of the apparent microscopic contact angles, $\theta_v$, $\theta_e$ and $\theta_c$ [39,69,88]. This relationship is given by Eq. (3), where we introduce the geometric factor $\Phi = \cos\theta_{in}$, which reflects the membrane affinity for the condensate and external phases[39,69].

Figure 3d, e shows the quantification of the geometric factor and the intrinsic contact angle for $K_{10}/D_{10}$ coacervates in contact with DOPC and DOPC:DOPS vesicles. For complete dewetting, $\Phi = 1$ and $\theta_{in} = 0°$, for partial wetting $-1 < \Phi < 1$ and $0° < \theta_{in} < 180°$, and for complete wetting $\Phi = -1$ and $\theta_{in} = 180°$. From the generic CG model simulations, we calculate the values of $\theta_{in}$ from several simulation snapshots across the trajectory and plot them in Fig. 3e (yellow stars). With carefully chosen peptide-lipid and lipid-lipid interaction parameters, the obtained values (with error bars) corroborate well the experimental values (hollow circles in Fig. 3e). To measure the angle, we took advantage of image processing tools namely, the Canny edge detection algorithm[65]; see the SI (Fig. S6) for details. For each system we have used 20 snapshots taken at a uniform interval from the trajectories, which provide 40 intrinsic contact angles per system.

As we will demonstrate in the following sections, the strong agreement between the experimental results and the simulations in describing condensate-membrane interactions not only reinforces the observed experimental findings but also provides a complementary, molecular-level perspective on these interfaces.

## Membrane dynamics at the contact region

Previous studies have reported that lipid dynamics can slow down upon contact with a condensate[37,87,89]. To determine whether this effect can also occur for the system explored here, we combined experimental and simulation data to compare lipid diffusion in the membrane region wetted by the coacervate with that in the bare membrane. Experimentally, we used FRAP, as shown in Fig. 4a. The results indicate that lipid diffusion in the wetted region is ~1.7 times slower than in the bare membrane. A similar reduction in diffusion is observed in vesicles composed of pure DOPC, as shown in the SI, Fig. S1a.

From the generic CG model simulations, we indirectly estimate the ratio of the lipid diffusion constants in the wetted and the bare regions, $D_{Wet}$ and $D_{Bare}$, respectively. A direct calculation of the diffusion constants from MD trajectories was not feasible because the residence time of the lipids at the contact region is too short to capture the long-timescale behavior of the mean-squared displacement. The velocity autocorrelation formalism cannot be applied as time in these CG models has no direct connection with real time.

To estimate the diffusion ratio, we measured the displacement of the lipid headgroups over a time interval $\Delta t^*$ for lipids in contact with the coacervate as well as those far from it. $\Delta t^*$ was varied from 0.25 to 3.0. For each value of $\Delta t^*$, the ratio of the squared total displacements provides an estimate of the ratio $D_{Bare}/D_{Wet}$. As shown in Fig. 4b, this ratio saturates near 1.35 for pure DOPC, 1.6 for 10% DOPS, and 1.8 for the 20% DOPS system. These ratios are in close agreement with those obtained from FRAP experiments (Figs. 4a and S1), further validating the model parameters. Since the reduced lipid mobility at the condensate-membrane interface may be linked to increased lipids ordering, as previously suggested[37,50], in the next section, we explore that possibility.

## Membrane packing and lipid ordering at the contact region

To assess whether lipid packing increases locally in our system, we labeled the membranes with a 0.5 mol% of LAURDAN and placed them in contact with unlabeled coacervates. LAURDAN is an environment-sensitive dye whose spectral properties respond to changes in lipid packing and hydration[90]. To evaluate its spectral shifts, we employed hyperspectral imaging combined with phasor analysis[70].

Hyperspectral imaging allows us to capture spectral information at every pixel. Spectral phasor analysis, which involves applying a Fourier transform to the hyperspectral data (see methods), provides a straightforward visualization and quantification of LAURDAN spectral changes. These methods have been previously employed to characterize membrane-condensate interactions[37] and details on their application to membrane systems can be found in ref. 70.

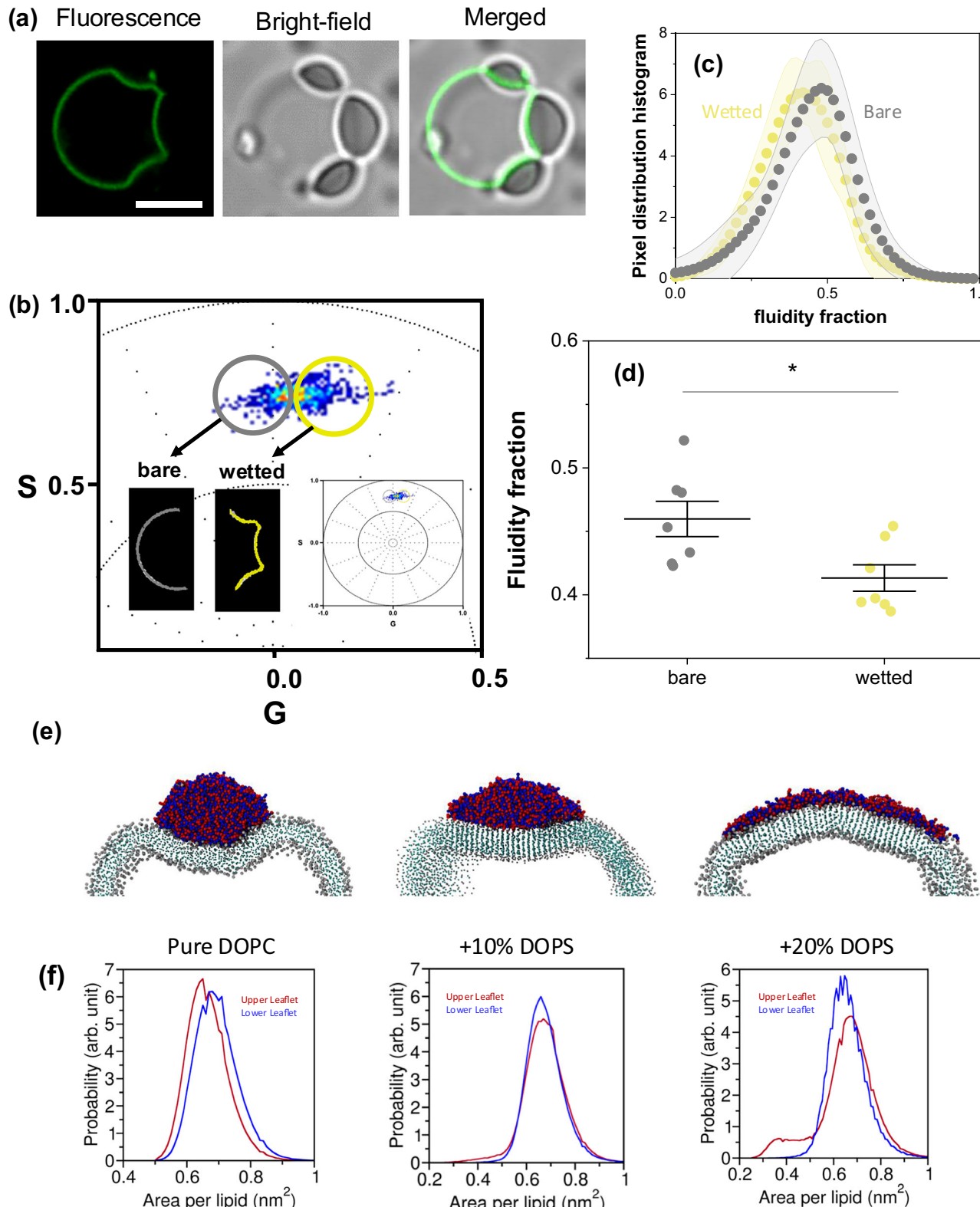

Figure 5a shows an example of a GUV composed of DOPC:DOPS (9:1) labeled with LAURDAN and in contact with three unlabeled $K_{10}/D_{10}$ coacervate droplets. The corresponding spectral phasor plot shown in Fig. 5b reveals distinct contributions from the wetted and bare membrane regions, indicating that LAURDAN is sensing two different local environments. Pixel distribution histograms for these membrane segments, obtained using the two-cursor analysis (see "Methods"), are shown in Fig. 5c, with their centers of mass plotted in Fig. 5d. These results clearly indicate that the membrane packing is enhanced in the membrane segment interacting with the coacervate compared to the bare membrane (SI, Fig. S1b, c for membranes made of pure DOPC).

When comparing the spectral phasors of vesicles composed of DOPC or DOPC:DOPS (9:1) in the absence of coacervates, we observed that incorporating 10 mol% of DOPS results in a decrease in lipid packing (SI, Fig. S2). Considering that DOPS is recruited to the

**Fig. 5 | Lipid packing and ordering at the coacervate-membrane interface.**
**a** GUVs composed of DOPC:DOPS (9:1) were labeled with 0.5 mol% LAURDAN and placed in contact with unlabeled $K_{10}/D_{10}$ coacervates. The images show an example of a GUV (green) wetted by three condensate droplets visible in bright field. Scale bar: 5 µm. **b** Spectral phasor plot for the vesicle displayed in (a). Using color cursor selection, the wetted (yellow) and bare (gray) membrane segments can be distinguished. The wetted membrane segment exhibits a blue shift, indicative of increased lipid packing. **c** Pixel distribution histograms for the wetted and bare membrane segments, presented as mean ± SD ($n = 8$). **d** Center of mass of histograms shown in (c). The wetted membrane segments display a reduced fluidity fraction (higher packing) compared to the bare segments (see SI, Fig. S9 for a

comparison among fluidity fractions of pure bilayers as well as bare and wetted segments). Individual data points are shown as circles, and the lines correspond to mean ± SD. Statistical significance was determined using ANOVA and Tukey post-test ($p < 0.05$). **e** Cross-sectional views of coacervates, from the generic model simulations, adsorbed on vesicles for three simulated systems showing orientational ordering of lipid tails at the coacervate-membrane contact region. With increased wetting, one observed stronger order in the tail region of the lipids. **f** Distributions of the area per lipid (APL) for the pure DOPC, 10% DOPS, and 20% DOPS systems, as obtained from MARTINI simulations. The leaflet in contact with the coacervate, termed the upper leaflet (red), shows a decrease in the APL values, i.e., increased packing.

membrane-coacervate interface (as shown in simulations, see Fig. 2c), and that it has a larger (and charged) headgroup than DOPC, one might expect that electrostatic repulsion would reduce lipid packing in the contact region. However, we observed the opposite effect: lipid packing increased in the wetted membrane segment, which agrees with the reduced lipid dynamics observed in Fig. 4. These results further support previous findings[48], suggesting a general consequence of coacervate-membrane interaction.

In the Cooke lipid simulations, we observed orientational ordering of the lipid tails at the coacervate-membrane interface for all systems except for the one corresponding to pure DOPC liposome (Fig. 5e). This ordering translates into tighter lipid packing in the contact region and provides enthalpic stabilization through increased interactions with the coacervate. The increased ordering is visible not only for the leaflet in contact with the condensate, but also for the underlying one (Fig. 5e). The observed local reorganization of membrane lipids associated with increased order is consistent with the membrane dehydration[48] and transmembrane coupling reported previously[87].

To further investigate lipid packing, we analyzed the APL distribution using MARTINI-3 simulations for both the upper leaflet (in contact with the coacervate) and the lower leaflet (Fig. 5f). For the pure DOPC system, we find a slightly left shifted APL distribution for the upper leaflet (Fig. 5f, *left panel*). For the 10% DOPS bilayer, the APL distribution for the upper leaflet shows an increase in lipid population with lower values of APL below 0.5 nm$^2$ (Fig. 5f, *middle panel*). Interestingly, in the 20% DOPS system, the APL distribution for the upper leaflet includes a significant population with values below 0.5 nm$^2$, indicative of a compact packing (Fig. 5f, *right panel*). A spatially resolved contour map of APL shows that the low APL values correspond to the "wetted" region of the bilayer (SI, Fig. S7). A component-wise analysis shows that the average APL values of the "free bilayer" systems are higher than that of the "wetted" systems. Moreover, DOPS lipids contribute to the decreased APL more significantly than DOPC lipids, consistent with the fact that DOPS heads are primarily in contact with the adsorbed coacervate (SI, Table S2).

While optical microscopy provides only limited insight into phase separation and the different emerging functions[21], the integration of advanced microscopy techniques with molecular dynamics simulations, as demonstrated here, offers a powerful approach to understanding the interactions between membrane-bound and membraneless moieties (as represented by the vesicles and coacervates here) at the nano- and molecular scale. A similar experimental strategy, employing fluorescence lifetime microscopy has shown that the micro-polarity of condensates is determinant for their structural organization such as the formation of multilayered condensates in organelles like the nucleolus[91]. In addition, measurements of fluorescence anisotropy can reveal molecular interactions within condensate during LLPS providing understanding in their material properties[92].

In this manner, the application of quantitative fluorescence microscopy techniques offers deeper insight into the diverse processes governing and involved in LLPS and the interactions with various organelles. The combination of these experimental approaches with molecular dynamics simulations enables a more comprehensive picture, extending resolution to the molecular scale.

## Conclusion
In this work, we have investigated the intricate interplay between coacervates and lipid membranes, an interaction fundamental to many cellular processes and the development of synthetic systems. By combining experimental observations of GUVs and $K_{10}/D_{10}$ coacervates with molecular dynamics simulations, we constructed an effective framework for elucidating the underlying mechanisms governing this complex phenomenon.

First, we successfully parameterized the coarse-grained model to accurately reproduce the fluid-elastic parameters observed in experiments, a critical step in establishing a reliable computational representation of the system. This allowed us to describe the wetting behavior of $K_{10}/D_{10}$ coacervates on membranes, characterized by the apparent contact angles and geometric factors at the microscale, as well as the intrinsic contact angle at the nanometric scale. When evaluating the membrane dynamic properties, our simulations further captured the experimentally observed mobility ratio between wetted and bare membrane regions, validating the approach of calibrating the coarse-grained model based on experimental contact angle parameters.

To elucidate the mechanism behind the reduced membrane fluidity at the coacervate interface, we performed hyperspectral imaging of LAURDAN combined with phasor analysis and demonstrated at the single-vesicle level that lipid packing increases in the wetted membrane segment compared to the bare membrane. Simulations provided molecular insight into this behavior, showing that ordering of lipids in the wetted region leads to a decrease in the APL and de-mixing. However, unlike the Cooke lipid model, the MARTINI bilayers do not exhibit significant lipid tail ordering. We believe this is due to the different nature of entropy-enthalpy interplay in these two models. The Cooke model has a very limited (if any) capacity to capture lipid conformational entropy. As a result, the system's behavior is primarily governed by effective enthalpic contributions. This leads to efficient lipid packing with well-aligned tails, thereby increasing the contact area between both lipid–lipid and protein–lipid interfaces. By contrast, MARTINI lipids possess significant tail conformational entropy, which prevents them from achieving such alignment even when the lipid head groups are concentrated beneath the coacervate. In other words, in MARTINI lipids, the gain in enthalpy from ordering is counterbalanced by the loss of tail conformational entropy, unlike in the Cooke model.

Overall, our multiscale approach demonstrated that experiments and simulations not only yield consistent information but also provide complementary perspectives, offering a richer view of these systems. This work represents a significant advancement in our ability to quantitatively describe coacervate-membrane interactions. These interactions are not only important for membrane remodeling processes, but also can be key in signaling events, since both the lipid de-mixing and reduction of lipid dynamics at the contact region point to the formation of organized structures with distinct properties.

Our results contribute to a deeper understanding of the factors influencing coacervate-membrane interactions, paving the way for future studies exploring the impact of different coacervate compositions, membrane properties, and environmental conditions on these interactions. For example, it is of great interest to study the effect of cholesterol, which plays major roles in modulating the phase behaviors of cell membranes and the

interaction between proteins and lipids[93,94]. Indeed, a thorough analysis of interactions between protein condensates and raft-like lipid domains will help better understand the molecular mechanisms of cell signaling and its regulation[8,54,58]. For computational studies in this context, while a MARTINI description of cholesterol is available, a model in the framework of the Cooke lipids needs to be developed and calibrated carefully against experimental observations, which is worthwhile as the simulation of phase behaviors of multi-component lipid membrane is still computationally demanding with the MARTINI model. Another topic of interest concerns the relative importance of IDRs and structured domains to condensate/membrane interactions, since many proteins interacting with lipid membranes feature both disordered and structured regions[95]. Our present study provides a sound basis for future studies in this exciting, rapidly developing area by offering a robust framework integrating experimental and computational approaches for the description of the structural and dynamic properties at the interface. Along this line, further integrating with non-linear spectroscopic techniques and atomistic simulations that provide complementary level information about the structure and dynamics at such interface will be valuable[47,96,97].

## Data availability

The relevant data from this work is available at: https://zenodo.org/records/15707442.

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

## Acknowledgements

A.M. acknowledges support from the Alexander von Humboldt foundation. Q.C. acknowledges the NSF grant, CHE-2154804. S.M. and Q.C. thank the computing resources provided by the Shared Computing Cluster (SCC) at Boston University. R.D. acknowledges the ComeInCell network funded by the European Union's Horizon Europe research and innovation program under the Marie Skłodowska-Curie grant agreement No. 101168939.

## Author contributions

Conceptualization: S.M., A.M., R.D., Q.C. Investigation: S.M., A.M. Formal analysis: S.M., A.M., R.D., Q.C. Funding acquisition: R.D., Q.C. Supervision: R.D., Q.C. Writing – original draft: S.M., A.M., R.D., Q.C. Writing – review & editing: S.M., A.M., R.D., Q.C.

## Funding

## Competing interests

The authors declare no competing interests.
