## [Transparent Peer Review file · Communications Chemistry]

Integrating experiments and simulations to unravel coacervate-membrane Interactions: Insights into de-mixing and morphology modulation

Corresponding Author: Professor Qiang Cui

Version 0:

Reviewer comments:

Reviewer #1

(Remarks to the Author)

In their intriguing manuscript, "Integrating experiments and simulations to unravel coacervate-membrane Interactions: Insights into de-mixing and morphology modulation", Mondal et al. use integrated experiments and simulations to investigate coacervate wetting of membranes. Their work reveals how membrane-condensate interactions tune the intrinsic contact angle of the membrane-condensate system and reduce the fluidity when in contact with the membrane. While I enjoyed the manuscript and found the synergy between simulation and experiment quite convincing, there are a few modifications I would recommend addressing before publication:

1. In the abstract, the authors claim "membrane-condensate affinity is tuned by the anionic lipid content"; however, isn't this likely specific to the peptide sequences they used? For example, a primarily negatively charged condensate would likely have a smaller contact angle as membrane charge becomes more negative?
2. Also in the abstract, the authors claim that "We find that the membrane in contact with the condensate displays a nearly two-fold reduced fluidity compared to the bare membrane. This is attributed to orientational ordering of lipid tails, resulting in decreased area per lipid." I definitely agree their data shows reduced fluidity and decreased area per lipid. However, can the authors be sure of the causal relationship? For example, it is the head groups of the membrane that are in contact with the condensate. Could head group - condensate interactions be behind both the decreased fluidity and the decreased area per lipid?
3. In an introductory paragraph, the authors state: "At the molecular scale, contact with a membrane can influence the conformational ensemble of intrinsically disordered proteins (IDPs), potentially altering their phase behavior. The single-chain properties (such as q -temperature and Boyle temperature) of an IDP were shown to be correlated with the condensate properties (such as the critical temperature). For example, fused in sarcoma (FUS) forms fibrillar β -sheet rich structures on phosphatidylserine membranes, whereas it forms entangled condensate on a phosphatidylglycerol membrane." I found these sentences misleading, as the FUS example relates to the first sentence ("At the molecular scale...") not the second sentence ("The single-chain properties...").
4. I found the explanation of Figure 2c ("Observation from MARTINI simulations...") to be unclear (Perhaps more explicitly draw the connection between the membrane composition and degree of wetting?).
5. The analysis of contact angle from simulation seems difficult to reproduce and reliant on user expertise. In particular, the first step ("First we find a suitable two-dimensional projection using Visual Molecular Dynamics (VMD) software and convert the colored image into a greyscale one (Step-1)") is done manually. Would it be possible to automate this entire process? Perhaps by performing a 2D projection along a plane including the vector through the center of mass of the condensate and the center of mass of the membrane?
6. Related to the previous comment, how many simulation snapshots was this analysis performed on? If this needed to be done manually for each snapshot, I worry about getting enough samples for good statistics.
7. Since the authors set up a new, generic CG model and changed MARTINI parameters, it may be useful to provide a small code repository with the input parameters, and at least one ready to run example for reproducibility.

Reviewer #2

(Remarks to the Author)

The authors present an integrated study combining experimental methods (confocal microscopy, hyperspectral imaging, FRAP) with CG and MARTINI simulations to investigate the interactions between K10/D10 coacervates and lipid membranes. They quantify membrane–condensate apparent and intrinsic contact angles and demonstrate consistency between experimental and computational results. A key finding is that coacervate adsorption reduces membrane fluidity by nearly two-fold, an effect attributed to lipid tail ordering and decreased area per lipid. Hyperspectral LAURDAN analysis provides experimental support for increased lipid packing in wetted regions, while simulations further reveal lipid de-mixing at the interface. Overall, the study establishes a multiscale framework for correlating structural and dynamic properties of condensate–membrane interactions.

Concerns and Points for Clarification

1. The manuscript refers to apparent contact angles, which vary depending on measurement conditions, and states that these were converted into intrinsic contact angles that represent material properties of the condensate/bilayer complex. Although three references are cited, the procedure for carrying out this conversion is not described. Since all of the simulation work relies on accurate intrinsic contact angles, the methodology for this crucial step should be explicitly presented. Moreover, snapshots from coarse-grained simulations, such as Figure 2b, show that the measured angles are not uniform, which seems inconsistent with the claim that the intrinsic contact angle is invariant for a specific condensate/bilayer system.
2. The condensate–vesicle mixture was sealed in a coverslip chamber for confocal imaging, but it is not clear whether the GUVs were immobilized on the coverslip surface or suspended in solution. If immobilized, the negatively charged coverslip may influence the wetting process of condensates on DOPC or DOPC/DOPS membranes. If suspended, the potential effects of laser-induced movement of GUVs, which commonly occurs during imaging, should be considered, especially with regard to the accuracy of FRAP measurements.
3. There are concerns regarding the interpretation of FRAP results. Figure 4a presents recovery curves for 10% DOPS vesicles, while Figure 4b shows diffusion constant ratios from CG simulations. Because the DOPS fraction affects lipid diffusion as shown in simulations, comparisons between experiments and coarse-grained models should be carried out at the same lipid composition (line 427, page 19). In addition, the manuscript equates the halftime of recovery from FRAP with diffusion constants, yet these quantities are not the same. A description of the quantitative relationship between halftime and diffusion constant, which depends on bleach spot geometry, should be included to enable a valid comparison. Another confusing point is the statement that lipid residence times at the contact region are too short to capture long-timescale behavior (line 419, page 19). Given that the contact region spans micrometer dimensions, does this imply that individual lipids move very rapidly in and out of these large regions? This interpretation seems counterintuitive and requires further explanation.
4. The analysis of area per lipid (APL) needs clarification. Figure 5f reports APL values for the inner and outer leaflets of MARTINI vesicles, and the authors conclude that the outer leaflet exhibits increased packing upon condensate adsorption. It is unclear whether this decrease was compared to the APL of the inner leaflet or to a control vesicle without condensates. The role of vesicle curvature in modulating inner and outer-leaflet APL values should also be addressed, along with the methodology used to assign APL to each leaflet in a curved vesicle. Furthermore, the manuscript states that MARTINI bilayers do not show significant lipid tail ordering, but this limitation is not sufficiently discussed. Since one of the main goals of the study is to demonstrate the predictive power of simulations for material properties of condensate/bilayer systems, a more detailed explanation of why lipid ordering is not reproduced in MARTINI simulations would strengthen the conclusions.

Version 1:

Reviewer comments:

Reviewer #1

(Remarks to the Author)

The authors did a good job responding to my comments. Two very small points:

1. A very small point, but I was not sufficiently clear about my confusion with the caption to Figure 2c. It was not due to the actual results of 2c, but rather the grammar in the sentence "Observation from MARTINI simulations: K10/D10-coacervate in contact with bilayer patches of three different composition: Wetting, negative curvature generation, and local lipid de-mixing upon the adsorption of a K10/D10-coacervate on a DOPC:DOPS=9:1 and DOPC:DOPS=8:2 bilayer patches." I think I understand what the authors intended based on the context, but the multiple colons (the latter of which seems to differentiate between the left and right portions of the figure without being labeled as such) makes the caption more complicated than it needs to be.

2. Might be nice to have the zenodo link in the text or SI (sorry if I missed it!)

Reviewer #2

(Remarks to the Author)

The authors have adequately addressed all of my previous comments and questions.

Authors' Response to Reviewers' comments:

Reviewer #1

In their intriguing manuscript, “Integrating experiments and simulations to unravel coacervate-membrane Interactions: Insights into de-mixing and morphology modulation”, Mondal et al. use integrated experiments and simulations to investigate coacervate wetting of membranes. Their work reveals how membrane-condensate interactions tune the intrinsic contact angle of the membrane-condensate system and reduce the fluidity when in contact with the membrane. While I enjoyed the manuscript and found the synergy between simulation and experiment quite convincing, there are a few modifications I would recommend addressing before publication:

Thank the Reviewer for the positive comments. We are pleased to learn that you found the manuscript intriguing and interesting, especially the synergy between theory and experiments, which was indeed a central aim of this study. Below, we address the points and concerns raised by the Reviewer. All the changes are incorporated in the revised manuscript.

1. In the abstract, the authors claim “membrane-condensate affinity is tuned by the anionic lipid content”; however, isn't this likely specific to the peptide sequences they used? For example, a primarily negatively charged condensate would likely have a smaller contact angle as membrane charge becomes more negative?

Reply: We agree with the Reviewer that the sequence (and composition) of the disordered polypeptides play an important role in the extent of wetting followed by membrane remodeling, as shown in a recent study by us [Mondal and Cui. *J. Phys. Chem. B* 2024, 128, 9, 2087–2099]. However, the quoted statement “membrane-condensate affinity ... content” is specific to the polyelectrolyte condensates in which an equimolecular positively (here K_{10}) and negatively (here D_{10}) charged polymers are present. Therefore, the net charge of the condensate is '0'. Since the lipids found in cell membranes are either neutral (such as PC lipids) or anionic (such as PG lipids), we mentioned the relationship between membrane-condensate affinity and anionic lipid content.

Nevertheless, as mentioned by the Reviewer, a net negatively charged condensate would exhibit the opposite trend in terms of bilayer wettability and contact angles, if the anionic lipid head groups are increased. Hence, we have rewritten the statement as follows in the revised manuscript:

“... we show that the membrane affinity of the K_{10}/D_{10} coacervate can be tuned by the anionic lipid content and quantified through the intrinsic contact angle ...”

2. Also in the abstract, the authors claim that “We find that the membrane in contact with the condensate displays a nearly two-fold reduced fluidity compared to the bare membrane. This is attributed to orientational ordering of lipid tails, resulting in decreased area per lipid.” I definitely agree their data shows reduced fluidity and decreased area per lipid. However, can the authors be sure of the causal relationship? For example, it is the head groups of the membrane that are in contact with the condensate. Could head group - condensate interactions be behind both the decreased fluidity and the decreased area per lipid?

Reply: We thank the Reviewer for this insightful comment. Indeed, the primary interactions in our system occur between the lipid head groups and the condensate, particularly given that the condensate components are highly charged and therefore interact minimally with the hydrophobic lipid tails.

The observed reduction in fluidity and decrease in area per lipid at the ‘wetted’ region can be attributed to enhanced lipid packing driven by favorable condensate-lipid head group interactions. This increased packing is enthalpically favorable and leads to an expansion of the contact area, which is, however, counterbalanced by the surface tension and internal cohesion of the condensate, preventing full spreading over the membrane. As a result, the system adopts a configuration with increased lipid packing in the wetted region at the cost of lipid tail conformational entropy.

According to our generic coarse-grained model, this increased packing correlates with a transition toward a more liquid-ordered phase. However, we note that this liquid-ordered phase is not explicitly observed in the MARTINI simulations, although they do confirm an increase in lipid packing. The hyperspectral LAURDAN analyses show lipid packing and dehydration of the polar head groups. However, LAURDAN measurements are indirect and it lacks atomistic resolution, therefore one needs the observations from simulations to interpret the experimental results.

Therefore, we have now changed that part in the abstract as: “**This is attributed to the crowding of lipids at the contact region, ...**”.

3. In an introductory paragraph, the authors state: “At the molecular scale, contact with a membrane can influence the conformational ensemble of intrinsically disordered proteins (IDPs), potentially altering their phase behavior. The single-chain properties (such as θ -temperature and Boyle temperature) of an IDP were shown to be correlated with the condensate properties (such as the critical temperature). For example, fused in sarcoma (FUS) forms fibrillar β -sheet rich structures on phosphatidylserine membranes, whereas it forms entangled condensate on a phosphatidylglycerol membrane.” I found these

sentences misleading, as the FUS example relates to the first sentence (“At the molecular scale...”) not the second sentence (“The single-chain properties...”).

Reply: Thank you for the comment. We agree that the discussion on the correlation between single-chain properties and condensate properties was not well aligned with the context highlighted by the Reviewer. Accordingly, we have removed that part in the revised manuscript.

4. I found the explanation of Figure 2c (“Observation from MARTINI simulations...”) to be unclear (Perhaps more explicitly draw the connection between the membrane composition and degree of wetting?).

Reply: We acknowledge the lack of clarity in the figure caption and discussion of Figure 2c. The connection between the membrane composition and the degree of wetting can be found in **Figure S8** (in the Supplemental Materials) where we plotted the number of contacts between different amino acids and lipid heads. The figures (and insets) show that the number of contacts increases with increasing DOPS content. Nevertheless, we now add another figure (Figure **S8.c**) showing the variation of the number of total contacts (between polyelectrolyte backbone beads and lipid heads) against the %DOPS content. A pair is said to be in contact if they are within a 5Å distance cut-off.

We have added a sentence in the revised manuscript (page 15, lines 367-368) and the addition to the Supplemental Materials is as follows:

Figure S8. (c) Variation of the total number of contacts ($r_{\text{cut}} = 5\text{\AA}$) between polyelectrolyte backbone beads and lipid heads against the % DOPS content, from MARTINI simulations.

5. The analysis of contact angle from simulation seems difficult to reproduce and reliant on user expertise. In particular, the first step (“First we find a suitable two-dimensional projection using Visual Molecular Dynamics (VMD) software and convert the colored image into a greyscale one (Step-1)”) is done manually. Would it be possible to automate

this entire process? Perhaps by performing a 2D projection along a plane including the vector through the center of mass of the condensate and the center of mass of the membrane?

Reply: We agree with the Reviewer that, at the current stage, the extraction of contact angles relies partly on user expertise. We have indeed made extensive efforts to automate this process. The initial steps, such as aligning the center-of-mass (COM) vector connecting the condensate and the vesicle with a box axis (say, Z), generating 2D projections perpendicular to the Z-axis, converting the images to greyscale, and applying Canny edge detection- can all be automated relatively straightforwardly. In fact, the 'suitable projections' were obtained by aligning the center-of-mass (COM) vector connecting the condensate and the vesicle with the Z-axis of the box.

However, the main challenge arises in defining smooth and physically meaningful interfaces around both the condensate and the vesicle. Since the simulations are performed at the nanometer scale (unlike experimental systems that operate at the micrometer scale), the extracted boundaries tend to be rugged and noisy. Automated smoothing often introduces significant errors in the measured intrinsic contact angle. Therefore, some degree of manual intervention remains necessary to ensure accuracy.

Nevertheless, we are actively exploring improved algorithms to reliably generate smooth envelopes from atomistic data and aim to achieve full automation in future. We have now discussed this challenge in the revised Supporting Information (updated Figure S6 caption).

6. Related to the previous comment, how many simulation snapshots was this analysis performed on? If this needed to be done manually for each snapshot, I worry about getting enough samples for good statistics.

Reply: We used 20 snapshots taken at a uniform interval from the trajectories. The 20 snapshots provide us with 40 intrinsic contact angles. We have added this information on page 17 (lines 403-405) of the revised text.

7. Since the authors set up a new, generic CG model and changed MARTINI parameters, it may be useful to provide a small code repository with the input parameters, and at least one ready to run example for reproducibility.

Reply: We have already uploaded the initial coordinates, re-parameterized MARTINI parameters, and the resultant trajectories in a repository. Here is the link: <https://zenodo.org/records/15707442>

Reviewer #2

The authors present an integrated study combining experimental methods (confocal microscopy, hyperspectral imaging, FRAP) with CG and MARTINI simulations to investigate the interactions between K10/D10 coacervates and lipid membranes. They quantify membrane–condensate apparent and intrinsic contact angles and demonstrate consistency between experimental and computational results. A key finding is that coacervate adsorption reduces membrane fluidity by nearly two-fold, an effect attributed to lipid tail ordering and decreased area per lipid. Hyperspectral LAURDAN analysis provides experimental support for increased lipid packing in wetted regions, while simulations further reveal lipid de-mixing at the interface. Overall, the study establishes a multiscale framework for correlating structural and dynamic properties of condensate–membrane interactions.

We sincerely thank the Reviewer for the insightful evaluation of our work. We have carefully considered all the comments and suggestions that have greatly helped us improve the clarity of the manuscript. Our detailed response is provided below.

1. The manuscript refers to apparent contact angles, which vary depending on measurement conditions, and states that these were converted into intrinsic contact angles that represent material properties of the condensate/bilayer complex. Although three references are cited, the procedure for carrying out this conversion is not described. Since all of the simulation work relies on accurate intrinsic contact angles, the methodology for this crucial step should be explicitly presented. Moreover, snapshots from coarse-grained simulations, such as Figure 2b, show that the measured angles are not uniform, which seems inconsistent with the claim that the intrinsic contact angle is invariant for a specific condensate/bilayer system.

Reply: The geometric factor calculated from the apparent contact angles as described in eq. 1 is a material property and only depends on the geometry of the condensate and vesicle systems. The intrinsic contact angle is related to the geometric factor through: $\Phi = \cos \theta_{in}$. We have added this sentence in the revised manuscript on page 7 (line 171).

We described the method to obtain the intrinsic contact angle from simulation trajectories in the Supporting Information section. From simulations, we do not obtain a single static structure but rather the full dynamic evolution of the system. Naturally, thermal fluctuations lead to variations in the instantaneous contact angles. These fluctuations are an intrinsic feature of the dynamics of the system. What is physically meaningful, therefore, is the ensemble-averaged contact angle (with its associated confidence interval), which should be compared with the experimental observable.

2. The condensate–vesicle mixture was sealed in a coverslip chamber for confocal imaging, but it is not clear whether the GUVs were immobilized on the coverslip surface or suspended in solution. If immobilized, the negatively charged coverslip may influence the wetting process of condensates on DOPC or DOPC/DOPS membranes. If suspended, the potential effects of laser-induced movement of GUVs, which commonly occurs during imaging, should be considered, especially with regard to the accuracy of FRAP measurements.

Reply: We thank the reviewer for raising this point. The GUVs were not immobilized on the coverslip through specific interactions but rather settled by gravity. This approach was chosen to avoid vesicle adhesion, which could alter the GUV geometry and thereby affect the accuracy of the contact angle measurement. To prevent the adhesion of GUVs to the glass surface, as well as the wetting of the coverslip by the condensates, we cleaned the coverslips with EtOH and water and then passivated them with a 2.5 mg/mL BSA solution. To facilitate stabilization by gravity, and thus imaging, vesicles are filled with a sucrose solution and then diluted in the (isotonic) condensate buffer containing glucose; this makes vesicles sink to the bottom of the chamber due to the density difference between the vesicle interior and exterior. In addition, the vesicles were further immobilized by the interaction with the condensate dense phase, which facilitated imaging as well as FRAP experiments **Figure S10** below shows that condensates do not wet the glass and remain spherical. When interacting with vesicles, condensates wet the membrane and deforms at the membrane-condensate interface.

We have now clarified this in the methods section (page 5) and included the figure in the SI.

Figure S10. (a) Oblique view of a 3D projection of $K_{10}D_{10}$ condensates. Condensates do not wet the glass surface and remain spherical. (b) Oblique view of a 3D projection of several condensates

interacting with a GUV. The condensates wet and change their shape at the membrane surface. The GUV does not wet the chamber bottom and gets immobilized by the interaction with the condensates. Scale bars are 5 μm .

3. (a) There are concerns regarding the interpretation of FRAP results. Figure 4a presents recovery curves for 10% DOPS vesicles, while Figure 4b shows diffusion constant ratios from CG simulations. Because the DOPS fraction affects lipid diffusion as shown in simulations, comparisons between experiments and coarse-grained models should be carried out at the same lipid composition (line 427, page 19).

Reply: We have now rewritten this paragraph:

“As shown in Figure 4b, this ratio saturates near 1.35 for pure DOPC, 1.6 for 10% DOPS and 1.8 for the 20% DOPS system. These ratios are in close agreement with those obtained from FRAP experiments (Figure 4a and S1), further validating the model parameters.”

3. (b) In addition, the manuscript equates the halftime of recovery from FRAP with diffusion constants, yet these quantities are not the same. A description of the quantitative relationship between halftime and diffusion constant, which depends on bleach spot geometry, should be included to enable a valid comparison.

Reply: We agree with the reviewer that this was not sufficiently clear in the manuscript. Now we have added the following to the FRAP methods section (pages 6, 148-157):

“the apparent diffusion constant can be obtained by the FRAP measurements through:

$$D_{app} = \frac{r_0^2 v}{4t_{1/2}}$$

where r_0 is the radius of the bleaching spot and v is a correction factor accounting for the difference between the defined size of bleaching spot and its real size (Ref. Giant Vesicle Book; DOI:10.1201/9781315152516). As the measurements conditions were the same for the wetted and bare parts of the vesicles (as well as for the different vesicles), the ratio of the $t_{1/2}$ for the wetted and bare parts of the membrane becomes equal to the ratio of the diffusion coefficients:

$$\frac{t_{1/2 \text{ wet}}}{t_{1/2 \text{ bare}}} = \frac{D_{app \text{ bare}}}{D_{app \text{ wet}}}$$

Since we compare ratios of diffusion coefficients rather than absolute values, the geometric and methodological constraints specific to each measurement are expected to largely cancel out.”

3. (c) Another confusing point is the statement that lipid residence times at the contact region are too short to capture long-timescale behavior (line 419, page 19). Given that the contact region spans micrometer dimensions, does this imply that individual lipids move very rapidly in and out of these large regions? This interpretation seems counterintuitive and requires further explanation.

Reply: The abovementioned statement was regarding the simulated systems where the contact region spans from a sub-nanometer to nanometer length scale, unlike the experimental systems where the contact region spans micrometer dimensions. We have clarified this in the revised manuscript (line 444).

4. (a) The analysis of area per lipid (APL) needs clarification. Figure 5f reports APL values for the inner and outer leaflets of MARTINI vesicles, and the authors conclude that the outer leaflet exhibits increased packing upon condensate adsorption. It is unclear whether this decrease was compared to the APL of the inner leaflet or to a control vesicle without condensates.

Reply: The conclusion regarding increased lipid packing is supported by the appearance of a shoulder in the area-per-lipid (APL) distribution of the upper leaflet (wetted by the condensate) at values lower than 0.5 nm^2 which was not present in the lower leaflet (NOT wetted by the condensate). To further sustain the finding, we have now compared the APL values of different lipid components of the same lipid bilayers but without the condensates (added in the SI as **Table S2**). These data clearly show that the average APL values of the ‘free bilayer’ systems are higher than that of the ‘wetted’ systems. Moreover, DOPS lipids contribute to the decreased APL more significantly than DOPC lipids, consistent with the fact that DOPS heads are primarily in contact with the adsorbed coacervate.

Table S2. Average values of the area per lipid (APL) of two different kinds of lipids with and without the coacervate adsorbed on it.

	Without adsorbed coacervate		With adsorbed coacervate	
	DOPC (nm^2)	DOPS (nm^2)	DOPC (nm^2)	DOPS (nm^2)
Pure DOPC (MARTINI)	0.660 ± 0.003	---	0.645 ± 0.003	---
10% DOPS (MARTINI)	0.658 ± 0.003	0.673 ± 0.005	0.657 ± 0.003	0.572 ± 0.004
20% DOPS (MARTINI)	0.655 ± 0.003	0.674 ± 0.005	0.648 ± 0.003	0.568 ± 0.007

4. (b) The role of vesicle curvature in modulating inner and outer-leaflet APL values should also be addressed, along with the methodology used to assign APL to each leaflet in a curved vesicle.

Reply: We used an open-source code, FATSLiM (doi:10.1093/bioinformatics/btw563), to calculate the APL values and the distributions. This code employs a Voronoi tessellation approach based on the positions of the lipid head groups, which properly accounts for membrane curvature. Consequently, a curved membrane does not artifactually yield lower APL values, even though, from a top-view projection, it might appear as if more lipids are confined within a smaller two-dimensional region. We have now clarified this in the revised manuscript (Methods section, Page 12 lines 304-306).

4. (c) Furthermore, the manuscript states that MARTINI bilayers do not show significant lipid tail ordering, but this limitation is not sufficiently discussed. Since one of the main goals of the study is to demonstrate the predictive power of simulations for material properties of condensate/bilayer systems, a more detailed explanation of why lipid ordering is not reproduced in MARTINI simulations would strengthen the conclusions.

Reply: Thanks for bringing up this important point. It is not yet entirely clear why MARTINI lipids do not show tail ordering (but Cooke model does). The Cooke model has a very limited (if any) capacity to capture lipid conformational entropy. As a result, the system's behavior is primarily governed by effective enthalpic contributions. This leads to efficient lipid packing with well-aligned tails, thereby increasing the contact area between both lipid-lipid and protein-lipid interfaces. By contrast, MARTINI lipids possess significant tail conformational entropy, which prevents them from achieving such alignment even when the lipid head groups are concentrated beneath the coacervate. In other words, in MARTINI lipids, the gain in enthalpy from ordering is counterbalanced by the loss of tail conformational entropy, unlike in the Cooke model. We have discussed this in the revised manuscript (pages 24-25).